# The role of albumin-corrected anion gap as a predictor of all-cause mortality in patients with Sepsis-AKI: A propensity score-matched cohort study

**Jian Liao**[1][⊙], **Xiao Xiao**[1][⊙], **Dingyu Lu**[2], **Maojuan Wang**[1], **Wei Huang**[1]*

**1** Intensive Care Unit, Deyang People's Hospital, Deyang, China, **2** Oncology Department, Deyang People's Hospital, Deyang, China

⊙ These authors contributed equally to this work.
* 1337539781@qq.com

## Abstract

### Background

The potential association between albumin-corrected anion gap at admission and prognosis in patients with sepsis-induced AKI remains uncertain. The purpose of this study was to explore the prognostic value of ACAG on mortality in patients with Sepsis-AKI.

### Methods

Data for this retrospective study were obtained from the MIMIC IV2.2 database. We used propensity score matching (PSM) and Cox proportional hazards regression analysis to evaluate the correlation between the ACAG and clinical outcomes in patients with Sepsis-AKI. Propensity score matching (PSM) analysis was conducted to minimize bias. Receiver operating characteristic curves were generated for albumin, AG, and ACAG, and comparisons of the areas under the ROC curves were made. Decision curve analysis (DCA) was carried out to assess the net benefit of ACAG.

### Results

According to the screening criteria, we identified a cohort of 2387 patients diagnosed with Sepsis-AKI. When comparing the normal-ACAG group(12–20 mmol/L) to the high-ACAG group(>20mmol/L)), it was found that the high-ACAG group exhibited longer stays in the ICU(5 days vs.4 days, P = 0.023) and higher hospital mortality rate(48.5% vs. 20.2%, P < 0.001). After matching, Cox regression analysis results showed that the high-ACAG group exhibited higher risk of hospital mortality (HR = 1.80, 95%CI: 1.27–2.56, P < 0.001). The area under the curve (AUC) values of

**Data availability statement:** All relevant data are within the paper and its Supporting Information files.

**Funding:** The author(s) received no specific funding for this work.

**Competing interests:** The authors have declared that no competing interests exist.

ACAG was 0.697 (after PSM), which was significantly higher than those of albumin or AG. ACAG also had the highest Youden's index and the largest net benefit range according to the decision curve analysis (DCA).

## Conclusion

Elevated serum ACAG (>20 mmol/L) is an independent risk factor for all-cause hospital mortality in patients with Sepsis-AKI. ACAG can be a new and easily acquired indicator that can provide new ideas for clinical practice.

## Introduction

Sepsis is defined as a life-threatening organ dysfunction caused by a dysregulated host response to infection. It is a complex clinical syndrome that results from the systemic inflammatory response to infection and can lead to tissue hypoperfusion and multiple organ failure if not promptly recognized and treated [1]. According to statistics, there are over 19 million sepsis patients worldwide annually, with approximately half of them being incurable. Sepsis causes approximately 6 million deaths each year [2]. There is limited understanding of the epidemiology of sepsis-associated acute kidney injury (Sepsis-AKI). It is clinically predicted that about 30% of sepsis patients will experience AKI. A pros pective cohort study, which observed 1,177 sepsis patients in various ICUs in Europe, indicated an AKI occurrence of 51% and a mortality rate of 41% [3]. Numerous studies have highlighted the difficulties in early detection and prediction of Sepsis-AKI using serum creatinine or urine output changes [4]. Several prognostic indicators have been identified for patients with Sepsis-AKI, such as IL-17A, IL-18, IL-33, HDL-C, KIM-1 [5–8]. However, the significance of these indicators continues to be a subject of debate.There is a critical need to investigate novel and accessible biomarkers that offer high sensitivity and specificity to promptly recognize and categorize patients at higher risk of severe AKI.

The anion gap (AG) serves as a crucial biochemical indicator for assessing the balance of acids and bases in the body. Research has demonstrated that AG not only offers insights into the severity of medical illnesses but also holds potential in forecasting patient prognosis [9–11].

Patients in the ICU often exhibit hypoalbuminemia, a condition in which albumin is crucial as the main contributor of unmeasured anions within the body. The charge associated with albumin may lead to inaccurately low results, masking raised anion gap levels and possibly leading to erroneous interpretations [12]. The albumin-corrected anion gap (ACAG) has been developed as a new biochemical marker to adjust for anion gap (AG) values, with the goal of providing a more precise indication of unmeasured anions [13]. Research has indicated that ACAG is linked to outcomes in individuals with sepsis [14] and kidney disorders [15]. In the field of cardiovascular disease, particularly in patients with myocardial infarction, the ACAG has been identified as a significant factor associated with 30-day mortality [16].

However, there is a lack of studies on the correlation between ACAG and the prognosis of patients with Sepsis-AKI. In order to address this gap, a retrospective cohort study was conducted to investigate the potential of the ACAG in predicting all-cause hospital mortality in patients with Sepsis-AKI.

## Methods

### Study population

The researchers conducted a retrospective observational study using data from the publicly accessible Medical Information Mart for Intensive Care IV (MIMIC-IV) database, which can be found at https://mimic.mit.edu. Specifically, the study examined the medical records of patients in the ICU at Beth Israel Deaconess Medical Center from 2008 to 2019. Ding yu Lu, as one of the authors, fulfilled the prerequisites to gain access to the database and undertook the task of data extraction (Record ID: 36142713). The patient cohort for this research comprised individuals with a confirmed diagnosis of Sepsis-AKI. The Third International Consensus Definitions for Sepsis and Septic Shock (Sepsis-3) was proposed recently [1]. Sepsi-AKI is usually defined as AKI in the presence of sepsis without other significant contributing factors explaining AKI or characterized by the simultaneous presence of both Sepsis-3 and Kidney Disease:Improving Global Outcomes (KDIGO) criteria [17,18]. AKI was diagnosed according to KDIGO criteria: (1) SCr increased by 0.3 mg/dL (or ≥ 26.5 μmol/L) within 48 h; or (2) increased by ≥ 1.5-fold from baseline within the prior 7 days; and/or (3) a decrease in urine output (UO) < 0.5 ml/kg/h for 6 h [19].

The use of the MIMIC-IV database was approved by the review committee of Massachusetts Institute of Technology and Beth Israel Deaconess Medical Center and patient's data were anonymized prior to publication. The Institutional Review Board of Deyang People's Hospital waived the requirement for written informed consent because of the study's retrospective design. All procedures were conducted in line with the pertinent guidelines and regulations.

The inclusion criteria were as follows: (1) patients with Sepsis-AKI during ICU admission; (2) aged ≥ 18 years; (3) albumin and anion gap could be calculated within 24 h from ICU admission; (4) first admission in ICU.

The exclusion criteria were as follows: (1) history of chronic kidney diseases or cancer; (2) length of ICU stay < 24h; (3) individuals who lacked adequate data on their first day of admission. Finally a total of 2387 patients formed the study cohort. (Fig.1).

### Data collection

To conduct the data extraction, we utilized PostgresSQL (version 13.7.2) software and Navicate Premium (version 16) tool by employing Structured Query Language (SQL). The extraction process prioritized four categories of potential variables: (1) demographic factors encompassing age, gender, weight and height; (2) comorbidities such as hypertension, diabetes, heart failure, acute myocardial infarction, COPD, pneumonia respiratory failure; (3) laboratory tests, including white blood cells (WBC), hemoglobin (HGB), platelet count (PLT), red blood cell distribution width (RDW), albumin, fast blood glugose (FBG); anion gap (AG); creatinine; lactate and so on. (4) We collected information on the use of mechanical ventilation (MV), continuous renal replacement therapy (CRRT), norepinephrine (NE) and furosemide. These laboratory variables were obtained solely during the initial 24-hour period following patient admission. In instances where there were multiple outcomes, the average measurement was employed. To mitigate any potential bias, variables containing missing values surpassing 20% were eliminated. To handle variables with less than 20% missing data, the research team employed the multiple imputation (missForest R) technique. ACAG were calculated using the formula: ACAG(mmol/l) = [4.4-{albumin (g/dl)}] *2.5+AG.

### Primary endpoints

The main outcome of this study was hospital all-cause mortality. Secondary outcome focused on length of stay in the ICU.

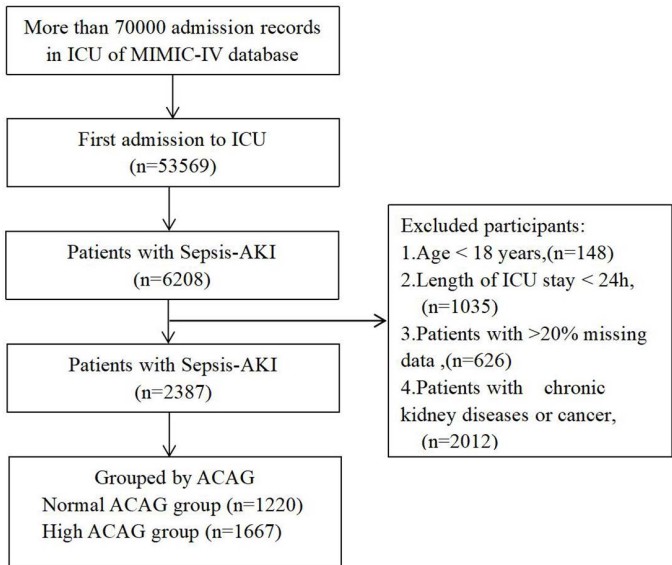

**Fig 1. Flow of included patients through the trial.**

## Statistical analysis

Continuous variables were presented as the mean ± SD or median and interquartile range (IQR). The comparison of continuous variables was performed using student's t-test and Wilcoxon rank-sum test, as appropriate, to compare the two groups. Categorical variables were expressed as numbers or percentages (%), and their analysis was implemented by means of chi-square test. Firstly, univariable Cox regression analysis was conducted to investigate the association between the ACAG and the endpoints. Variables that showed clinical significance and had a significance level of P < 0.05 were included in the multivariable Cox proportional hazards model. Secondly, the R language 'psmatching' extension package (V.3.04) was used to develop a matched group with a ratio of 1:1. All the matching parameters were predefined and previously described. The propensity matching score was calculated using logistic regression. The matching was done through the nearest neighbour method with caliper width limited to 0.05. The quality of the match was determined using standardised mean differences (SMDs). After matching, to investigate the association between ACAG and the risk of hospital all-cause mortality in patients with Sepsis-AKI, restricted three times spline plots were used. Thirdly, Kaplan-Meier survival analysis was used to assess the incidence rate of primary outcome events in different groups based on the ACAG, the log-rank test was employed to examine any observed disparities. In addition, Z test was used to compare the predictive value of Albumin, AG and ACAG by comparing the area under curves (AUC) of the receiver operating characteristic curves (ROC). Finally, we performed the decision curve analysis (DCA), to explore the net benefits between ACAG and in-hospital mortality in intensive care patients with Sepsis-AKI.

The data analyses were conducted using R software (version 4.2.2). For all analyses, a 2-side P < 0.05 was considered statistically significant.

## Results

### Patient characteristics

The study enrolled a total of 2387 critically ill patients with Sepsis-AKI from the MIMIC-IV database. Referring to previous study (15), patients were divided into normal-ACAG group (12–20 mmol/L, n = 1220) and high-ACAG group (>20

mmol/L,n = 1167). Before matching, significant differences in variables were observed between the normal and high-ACAG groups (P < 0.05), except for the Age, gender, BMI, platelet count, HGB, Ca, FBG, and the prevalence of commorbidities. Compared with the normal-ACAG group, it was found that the high-ACAG group demonstrated more pronounced renal function impairment, a greater prevalence of AKI stage 3(60.5% vs. 46.7%, P < 0.001), and a higher utilization of CRRT(23.7% vs. 11.1%, P < 0.001). Moreover, patients in high-ACAG group exhibited longer stays in the ICU(5 days vs. 4 days, P = 0.023) and higher hospital mortality rate (48.5% vs. 20.2%, P < 0.001). The researchers implemented PSM in order to balance any potential confounding factors that may have influenced the results. This meticulous matching process yielded a final sample of 910 patients between the normal- and high-ACAG groups. The study obtained a well-balanced matching cohort. Moreover, the analysis after matching revealed that all variables exhibited p-values > 0.05 except for the outcome measures (Table 1).

### Comparisons of prognosis between the normal- and high-ACAG groups

The prediction of hospital mortality was analyzed using univariate and multivariate Cox proportional hazards regression. Factors with P < 0.05 in univariate analysis were included in multivariate analysis, Cox regression analysis results showed that the high-ACAG group exhibited higher risk of hospital mortality (HR = 1.62, 95%CI: 1.31–2.01, P < 0.001) (Table 2).

After matching, multivariate Cox proportional hazards regression was used to reassess the relationship between different ACAG-group and the risk of mortality.The results showed that the high-ACAG group also exhibited higher risk of hospital mortality (HR = 1.80, 95%CI: 1.27–2.56, P < 0.001) (Table 3).

RCS showed that there was a non-linear relationship between ACAG at ICU admission and the risk of all-cause mortality during hospitalization in patients with Sepsis-AKI (P for non-linearity <0.001). When ACAG was 20.28 mmol/L, its HR was 1(Fig 2).

### Kaplan-Meier survival curve analysis

Before or after matching, the Kaplan-Meier survival curves in Fig 3, showed that compared with the normal-ACAGgroup, the cumulative survival rate of patients with Sepsis-AKI was significantly lower (P < 0.001) in the high-ACAG group during hospitalization.

### Predictive value of albumin, AG and ACAG for in-hospital mortality (after PSM)

ROC curves were performed to evaluate the predictive value of albumin, AG, and ACAG for in-hospital mortality in patients with Sepsis-AKI. After PSM, the AUC of albumin, AG, and ACAG were 0.551, 0.581, and 0.697, respectively (Fig. 4). ACAG had the highest sensitivity at 63.8% and the highest Youden's index at 0.3149, while AG had the highest specificity at 73.7% (Table 4).

### Comparison of decision curves

The interpretation of the decision curves is similar to that of ROC curves. A larger area under the curve indicates a greater net clinical benefit of the model. After propensity score matching (PSM), the net benefit ranges under the decision curves were ACAG, AG, and albumin in descending order (Fig 5). This suggests that ACAG was the optimal prognostic factor among the three index.

## Discussion

To the best of our knowledge, this study represents the first investigation into the association between the ACAG and all-cause hospital mortality in critically ill patients with Sepsis-AKI. Notably, through propensity score matching analysis, we adjusted for the baseline differences between the normal and high-ACAG group, and identified a significant correlation

**Table 1. Baseline characteristics between normal- and high-ACAG group.**

| Variables | Unmatched | | | Matched | | |
|---|---|---|---|---|---|---|
| | Normal-group(≤ 20) N = 1220 | High-group (> 20) N = 1167 | P SMD | Normal-group(≤ 20) N = 455 | High-group (> 20) N = 455 | P SMD |
| Age, year | 66 ± 17 | 66 ± 16 | 0.740 0.040 | 67 ± 16 | 66 ± 16 | 0.260 0.041 |
| BMI,% | 28.1 (24.6, 32.2) | 28.3(24.6, 33.0) | 0.124 0.048 | 28.0 (23.4, 32.7) | 28. 2(24, 32.5) | 0.812 0.022 |
| Male,n(%) | 702 (57.5%) | 647 (55.4%) | 0.301 0.030 | 238 (52.3%) | 247 (54.3%) | 0.550 0.014 |
| Laboratory tests | | | | | | |
| WBC,K/uL | 13 (8 19 ) | 15 (9 22 ) | <0.001 0.221 | 14 (9 20 ) | 15 (9 22 ) | 0.586 0.013 |
| Platelet, K/uL | 169 (109, 244) | 168 (100, 257) | 0.740 0.840 | 175 (113, 260) | 166 (103, 270) | 0.525 0.026 |
| HGB,g/dL | 10.26±2.20 | 10.43±2.35 | 0.069 0.080 | 10.38±2.26 | 10.15±2.22 | 0.125 0.018 |
| RDW,% | 15.69±2.52 | 16.13±2.78 | <0.001 0.215 | 15.87±2.72 | 16.09±2.73 | 0.234 0.039 |
| Albumin, g/dL | 2.75±0.53 | 2.65±0.59 | <0.001 0.341 | 2.67±0.55 | 2.67±0.55 | 0.921 0.069 |
| AG,mmol/L | 14.1±3.4 | 19.9±5.0 | <0.001 0.435 | 17.8±3.7 | 18.9±6.6 | 0.526 0.016 |
| ACAG,mmol/L | 17.3±2.0 | 24.7±4.4 | <0.001 0.361 | 18.2±1.6 | 24.3±5.7 | <0.001 0.539 |
| Na,mmol/L | 138±6 | 138±7 | 0.052 0.034 | 138±7 | 138±6 | 0.499 0.031 |
| K,mmol/L | 4.11±0.76 | 4.33±0.92 | <0.001 0.472 | 4.16±0.85 | 4.23±0.86 | 0.209 0.028 |
| Ca,mmol/L | 7.82±0.88 | 7.84±0.96 | 0.548 0.027 | 7.87±0.92 | 7.81±0.93 | 0.323 0.042 |
| FBG,mg/dL | 129 (104, 162) | 131 (101, 188) | 0.114 0.094 | 131 (104, 166) | 129 (101, 185) | 0.967 0.058 |
| Lactate,mg/dL | 1.90 (1.40, 2.80) | 2.80 (1.80, 4.70) | <0.001 0.309 | 2.10 (1.50, 3.20) | 2.50(1.60, 4.60) | 0.081 0.027 |
| INR | 1.40 (1.20, 1.80) | 1.50 (1.30, 2.00) | <0.001 0.461 | 1.50 (1.20, 1.88) | 1.50 (1.30,2.00) | 0.052 0.099 |
| ALT,U/L | 35 (19, 75) | 47 (22, 131) | <0.001 0.529 | 39 (20, 87) | 41 (21, 94) | 0.291 0.077 |
| AST,U/L | 54 (27, 107) | 82 (37, 218) | <0.001 0.515 | 59 (29, 135) | 72 (32, 154) | 0.066 0.026 |
| BUN,mg/dL | 27 (19, 40) | 38 (24, 57) | <0.001 0.239 | 31 (21, 46) | 34 (22, 52) | 0.065 0.037 |
| Creatinine,mg/dL | 1.30 (1.00, 1.70) | 1.80 (1.20, 2.70) | <0.001 0.338 | 1.50 (1.10, 2.00) | 1.60(1.20, 2.40) | 0.051 0.031 |
| SOFA | 7.5±3.7 | 9.7±4.4 | <0.001 0.183 | 8.3±3.8 | 8.9±4.4 | 0.227 0.052 |
| APSIII | 59±23 | 74±25 | <0.001 0.175 | 66±23 | 68±25 | 0.213 0.080 |
| SAPSII | 43±15 | 52±16 | <0.001 0.462 | 48±14 | 49±16 | 0.205 0.056 |
| OASIS | 35±9 | 40±9 | <0.001 0.291 | 37±9 | 38±9 | 0.312 0.047 |
| Commorbidities | | | | | | |
| Hypertension,n(%) | 628 (51.5%) | 611 (52.4%) | 0.667 0.048 | 251 (55.2%) | 237 (52.1%) | 0.352 |
| Diabetes,n(%) | 319 (26.1%) | 343 (29.4%) | 0.077 0.088 | 132 (29.0%) | 135 (29.7%) | 0.827 0.067 |
| Heart failure,n(%) | 312 (25.6%) | 302 (25.9%) | 0.865 0.044 | 123 (27.0%) | 122 (26.8%) | 0.940 0.023 |
| AMI,n(%) | 90 (7.4%) | 111 (9.5%) | 0.060 0.017 | 37 (8.1%) | 37 (8.1%) | 0.990 0.026 |
| Pneumonia,n(%) | 575 (47.1%) | 522 (44.7%) | 0.239 0.052 | 209 (45.9%) | 197 (43.3%) | 0.424 0.062 |
| COPD,n(%) | 104 (8.5%) | 98 (8.4%) | 0.911 0.030 | 51 (11.2%) | 38 (8.4%) | 0.147 0.011 |
| AKI stage | | | <0.001 0.325 | | | 0.624 0.035 |
| 1 | 172 (15.4%) | 103 (10.6%) | | 56 (12.3%) | 52 (11.4%) | |
| 2 | 425(37.9%) | 337 (28.9%) | | 158 (34.7%) | 135 (29.7%) | |
| 3 | 623 (46.7%) | 727 (60.5%) | | 241 (53.0%) | 268 (58.9%) | |
| CRRT,n(%) | 135 (11.1%) | 276 (23.7%) | <0.001 0.452 | 74 (16.3%) | 72 (15.8%) | 0.857 0.052 |
| Ventilation,n(%) | 572 (46.9%) | 697 (59.7%) | <0.001 0.399 | 254 (55.8%) | 243 (53.4%) | 0.464 0.047 |
| NE,n(%) | 848 (69.5%) | 916 (78.5%) | <0.001 0.275 | 342 (75.2%) | 340 (74.7%) | 0.878 0.029 |
| Furosemide,n(%) | 563 (46.1%) | 534 (45.8%) | 0.849 0.016 | 229 (50.3%) | 249 (54.7%) | 0.184 0.051 |
| Hospital mortality,n(%) | 247 (20.2%) | 566 (48.5%) | <0.001 0.583 | 86 (18.9%) | 219 (48.1%) | <0.001 0.572 |

*(Continued)*

**Table 1.** (Continued)

| Variables | Unmatched | | | Matched | | |
|---|---|---|---|---|---|---|
| | Normal-group(≤ 20) N = 1220 | High-group (> 20) N = 1167 | P SMD | Normal-group(≤ 20) N = 455 | High-group (> 20) N = 455 | P SMD |
| Hospital_Los,Day | 13 (7 24 ) | 11 (6 21 ) | <0.001 0.492 | 15 (8, 26) | 11 (6 20 ) | <0.001 0.485 |
| ICU_Los,Day | 4 (2 9 ) | 5 (2 10 ) | 0.023 0.275 | 5 (2 12 ) | 4 (2 9 ) | <0.001 0.673 |

Abbreviation: BMI: Body mass index; WBC: White blood cell; RDW: Red blood cell distribution width; FBG:Fast Blood Glugose; AG:Anion gap;ACAG:-Albumin Corrected Anion Gap; INR:, international normalized ratio; ALT: glutamic-pyruvicntransaminase; AST: aspartate amino transferase;BUN:blood urea nitrogen; AMI, acute myocardial infarction;AKI:Acute kidney injury; CRRT: Continuous Renal Replacement Therapy; NE:norepinephrine; SOFA: Sequential organ failure assessment; APSIII, Acute Physiology Score III; SAPSII: Simplifed Acute Physiological Score II; OASIS:Oxford acute severity of illness score; Hospital_Los: Hospital Length of Stay;ICU-Los: ICU Length of Stay.

between ACAG and in-hospital mortality of Sepsis-AKI. Whether before PSM or after PSM, analysis of multivariate Cox proportional hazards regression showed that elevated ACAG (>20 mmol/L) was an independent risk factor of increased all-cause mortality during hospitalization in patients with Sepsis-AKI. The Kaplan-Meier survival curve indicated a significantly lower cumulative survival rate during hospitalization in the high-ACAG group. In our study, we conducted an analysis of ROC curves to determine the predictive value of three factors: ACAG, AG, and albumin. Our results indicated that ACAG was the most predictive factor, followed by AG and albumin. The findings suggest that ACAG may be a more reliable indicator in predicting the outcome compared to the other two factors. ACAG > 20.89 mmol/l (after PSM) can predict the risk of hospital mortality in patients with Sepsis-AKI.

Anion gap (AG) is frequently utilized in the diagnosis of acid-base imbalances, particularly for distinguishing the underlying cause and type of metabolic acidosis. Previous research has revealed that AG levels are correlated with a negative prognosis and function as a standalone indicator for mortality during hospitalization in individuals experiencing cardiac arrest [9]. A high anion gap is often associated with severe clinical prognosis, especially in patients with sepsis [14]. Patients with sepsis often have hyperlactatemia and metabolic acidosis, resulting in increased AG, but patients with sepsis are often accompanied by hypoalbuminemia. Since albumin is negatively charged that will lead to a decrease in AG,hypo-albuminemia may lead to false negative AG results, affecting the accurate judgment of the results. Therefore, the ACAG was used to more accurately reflect the presence of unmeasured anions by adjusting AG based on albumin levels. Zhang and colleagues noted that an AG measurement of ≥ 11.15 mmol/L was connected to heart-related incidents during a 1-year observation period in patients with acute coronary syndrome [20]. Additionally, AG was identified as an autonomous risk factor for anticipating mortality during hospitalization in patients facing acute ischemic stroke [21]. Furthermore, numerous investigations have indicated that heightened AG levels are also linked to a heightened likelihood of heart-related incidents or mortality in illnesses like kidney disorders [22], sepsis [14,23], acute pancreatitis [11,24], disseminated intravascular coagulation [25], aortic aneurysm [10], and other related conditions. However, the presence of albumin's charge can sometimes lead to a falsely normal anion gap [12]. Our study found that only slight modifications to AG significantly enhanced its predictive accuracy. We posit that ACAG has the capacity to supplant AG in the clinical situations.

Because of the restrictions of AG as a diagnostic instrument and its vulnerability to imprecise measurements, there has been an increasing necessity for a fresh indicator that can offer more precise forecasts of patient results. To tackle this problem, the notion of ACAG was created. In recent times, ACAG has been employed to anticipate the outlook of different illnesses, including individuals with acute kidney injury and patients undergoing coronary artery bypass graft [15,26]. The ACAG has been recognized as an independent risk factor associated with mortality, and its cut-off values differ among various diseases. For instance, ICU patients suffering from sepsis have a cut-off value exceeding 21.25 mmol/L, whereas for individuals experiencing cardiac arrest, the cut-off is set at more than 20 mmol/L [14]. Analysis of the ROC curve in

**Table 2. All-cause mortality during hospitalization between two groups before PSM.**

| Characteristic | Univariable | | | Multivariable | | |
|---|---|---|---|---|---|---|
| | HR | 95%CI | P | HR | 95%CI | P |
| Age | 1.02 | 1.01, 1.02 | <0.001 | 1.02 | 1.02, 1.03 | <0.001 |
| Male | 0.81 | 0.71, 0.93 | 0.003 | 0.85 | 0.74, 0.99 | 0.031 |
| BMI | 0.99 | 0.99, 1.00 | 0.137 | | | |
| WBC | 1.01 | 1.00, 1.01 | 0.013 | 1.00 | 1.00, 1.01 | 0.202 |
| Platelet | 1.00 | 1.00, 1.00 | <0.001 | 1.00 | 1.00, 1.00 | 0.004 |
| HGB | 0.97 | 0.94, 1.00 | 0.032 | 1.00 | 0.97, 1.03 | 0.950 |
| RDW | 1.08 | 1.06, 1.10 | <0.001 | 1.05 | 1.02, 1.08 | <0.001 |
| Albumin | 0.97 | 0.87, 1.09 | 0.596 | | | |
| AG | 1.06 | 1.05, 1.07 | <0.001 | 0.91 | 0.89, 0.94 | <0.001 |
| ACAG | 1.09 | 1.08, 1.10 | <0.001 | 1.10 | 1.07, 1.13 | <0.001 |
| High ACAG(> 20)† | 2.69 | 2.31, 3.12 | <0.001 | 1.62 | 1.31, 2.01 | <0.001 |
| Na | 1.00 | 0.99, 1.01 | 0.864 | | | |
| K | 1.14 | 1.06, 1.23 | <0.001 | 1.09 | 1.00, 1.19 | 0.043 |
| Ca | 1.05 | 0.98, 1.12 | 0.209 | | | |
| FBG | 1.00 | 1.00, 1.00 | 0.714 | | | |
| Lac | 1.13 | 1.11, 1.16 | <0.001 | 1.04 | 1.01, 1.08 | 0.011 |
| INR | 1.16 | 1.12, 1.21 | <0.001 | 1.09 | 1.03, 1.14 | <0.001 |
| ALT | 1.00 | 1.00, 1.00 | <0.001 | 1.00 | 1.00, 1.00 | 0.449 |
| AST | 1.00 | 1.00, 1.00 | <0.001 | 1.00 | 1.00, 1.00 | 0.164 |
| BUN | 1.01 | 1.00, 1.01 | <0.001 | 1.00 | 1.00, 1.00 | 0.895 |
| Creatinine | 1.04 | 1.00, 1.09 | 0.057 | | | |
| Hypertension | 0.93 | 0.81, 1.07 | 0.314 | | | |
| Diabetes | 0.89 | 0.76, 1.04 | 0.133 | | | |
| Heart failure | 1.03 | 0.88, 1.21 | 0.683 | | | |
| AMI | 1.16 | 0.92, 1.46 | 0.215 | | | |
| Pneumonia | 1.00 | 0.87, 1.15 | 0.988 | | | |
| COPD | 1.02 | 0.80, 1.30 | 0.872 | | | |
| AKI stage‡ | | | | | | |
| 2 | 1.45 | 1.02, 2.04 | 0.037 | 0.66 | 0.54, 0.80 | <0.001 |
| 3 | 2.92 | 2.12, 4.03 | <0.001 | 2.41 | 2.12, 2.78 | <0.001 |
| CRRT | 1.49 | 1.28, 1.73 | <0.001 | 0.80 | 0.67, 0.95 | 0.013 |
| Ventilation | 1.32 | 1.13, 1.53 | <0.001 | 1.03 | 0.86, 1.24 | 0.732 |
| SOFA | 1.10 | 1.08, 1.12 | <0.001 | 1.02 | 0.99, 1.05 | 0.257 |
| APSIII | 1.02 | 1.01, 1.02 | <0.001 | 1.01 | 1.01, 1.02 | <0.001 |
| SAPSII | 1.03 | 1.02, 1.03 | <0.001 | 0.99 | 0.99, 1.00 | 0.135 |
| OASIS | 1.04 | 1.03, 1.05 | <0.001 | 1.00 | 0.99, 1.02 | 0.565 |
| NE | 1.97 | 1.60, 2.44 | <0.001 | 1.46 | 1.16, 1.84 | 0.001 |
| Furosemide | 0.73 | 0.64, 0.84 | <0.001 | 0.57 | 0.49, 0.66 | <0.001 |

HR = Hazard Ratio CI = Confidence Interval

Abbreviation: BMI: Body mass index; WBC: White blood cell; RDW: Red blood cell distribution width; FBG:Fast Blood Glugose; AG:Anion gap;ACAG:Albumin Corrected Anion Gap; INR:, international normalized ratio;ALT:glutamic-pyruvic transaminase;AST:aspartate aminotransferase;BUN:blood urea nitrogen; AMI:acute myocardial infarction;AKI:Acute kidney injury; CRRT:Continuous Renal Replacement Therapy; NE:norepinephrine; SOFA: Sequential organ failure assessment;APSIII, Acute Physiology Score III;SAPSII: Simplifed Acute Physiological Score II;OASIS:Oxford acute severity of illness score.

† Compared with the normal ACAG group (≤20).

‡ Compared with the Stage 1.

**Table 3. All-cause mortality during hospitalization between two groups after PSM.**

| Characteristic | Univariable | | | Multivariable | | |
|---|---|---|---|---|---|---|
| | HR | 95%CI | P | HR | 95%CI | P |
| Age | 1.02 | 1.01, 1.02 | <0.001 | 1.02 | 1.01, 1.03 | <0.001 |
| Male | 0.86 | 0.69, 1.08 | 0.196 | | | |
| BMI | 1.00 | 0.98, 1.01 | 0.498 | | | |
| WBC | 1.01 | 1.00, 1.01 | 0.256 | | | |
| Platelet | 1.00 | 1.00, 1.00 | 0.008 | | | |
| HGB | 0.94 | 0.90, 0.99 | 0.022 | 1.00 | 0.94, 1.06 | 0.942 |
| RDW | 1.08 | 1.04, 1.12 | <0.001 | 1.03 | 0.98, 1.08 | 0.209 |
| Albumin,g/dL | 1.11 | 0.92, 1.33 | 0.292 | | | |
| AG | 1.04 | 1.02, 1.06 | <0.001 | 0.89 | 0.85, 0.92 | <0.001 |
| ACAG | 1.10 | 1.08, 1.11 | <0.001 | 1.14 | 1.09, 1.19 | <0.001 |
| High ACAG(> 20)† | 3.24 | 2.52, 4.16 | <0.001 | 1.80 | 1.27, 2.56 | <0.001 |
| Na | 1.00 | 0.99, 1.02 | 0.719 | | | |
| K | 1.14 | 1.01, 1.29 | 0.034 | 1.13 | 0.96, 1.32 | 0.139 |
| Ca | 1.02 | 0.91, 1.15 | 0.703 | | | |
| FBG | 1.00 | 1.00, 1.00 | 0.505 | | | |
| Lactate | 1.12 | 1.09, 1.15 | <0.001 | 1.06 | 1.00, 1.13 | 0.053 |
| INR | 1.23 | 1.14, 1.32 | <0.001 | 1.10 | 0.97, 1.24 | 0.155 |
| ALT | 1.00 | 1.00, 1.00 | 0.083 | | | |
| AST | 1.00 | 1.00, 1.00 | 0.028 | 1.00 | 1.00, 1.00 | 0.968 |
| BUN | 1.00 | 1.00, 1.01 | 0.112 | | | |
| Creatinine | 1.04 | 0.97, 1.13 | 0.252 | | | |
| Hypertension | 0.91 | 0.72, 1.14 | 0.393 | | | |
| Diabetes | 0.79 | 0.61, 1.02 | 0.068 | | | |
| Heart failure | 1.27 | 1.00, 1.61 | 0.052 | | | |
| AMI | 0.94 | 0.63, 1.39 | 0.741 | | | |
| Pneumonia | 1.03 | 0.82, 1.29 | 0.782 | | | |
| COPD | 1.00 | 0.69, 1.45 | 0.996 | | | |
| AKI stage‡ | | | | | | |
| 2 | 1.43 | 0.77, 2.66 | 0.253 | | | |
| 3 | 2.81 | 1.57, 5.05 | <0.001 | 2.42 | 2.06,2.88 | 0.016 |
| CRRT | 1.48 | 1.14, 1.90 | 0.003 | 0.81 | 0.59, 1.11 | 0.194 |
| Ventilation | 1.35 | 1.06, 1.73 | 0.016 | 1.21 | 0.89, 1.65 | 0.224 |
| SOFA | 1.09 | 1.07, 1.12 | <0.001 | 1.02 | 0.97, 1.07 | 0.407 |
| APSIII | 1.01 | 1.01, 1.02 | <0.001 | 1.01 | 1.00, 1.02 | 0.058 |
| SAPSII | 1.02 | 1.02, 1.03 | <0.001 | 0.99 | 0.98, 1.01 | 0.290 |
| OASIS | 1.03 | 1.02, 1.05 | <0.001 | 1.00 | 0.98, 1.02 | 0.761 |
| NE | 2.09 | 1.47, 2.97 | <0.001 | 1.30 | 0.87, 1.94 | 0.197 |
| Furosemide | 1.30 | 1.04, 1.63 | 0.022 | 1.34 | 1.01, 1.66 | 0.081 |

HR = Hazard Ratio CI = Confidence Interval

Abbreviation: BMI: Body mass index; WBC: White blood cell; RDW: Red blood cell distribution width; FBG:Fast Blood Glugose; AG:Anion gap;ACAG:-Albumin Corrected Anion Gap; INR:, international normalized ratio;ALT:glutamic-pyruvic transaminase;AST:aspartate aminotransferase;BUN:blood urea nitrogen; AMI:acute myocardial infarction;AKI:Acute kidney injury; CRRT:Continuous Renal Replacement Therapy; NE:norepinephrine; SOFA: Sequential organ failure assessment;APSIII, Acute Physiology Score III;SAPSII: Simplifed Acute Physiological Score II; OASIS:Oxford acute severity of illness score.

† Compared with the normal ACAG group (<20)

‡ Compared with the Stage 1

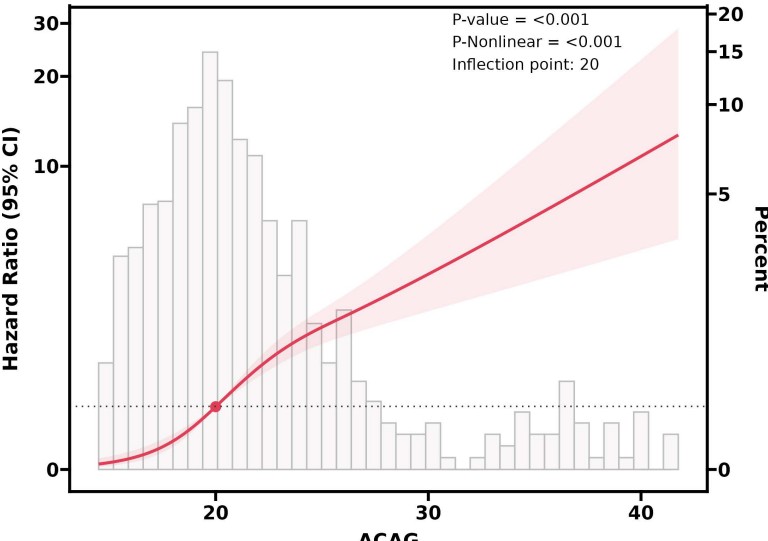

**Fig 2. Restricted cubic spline curve for the ACAG hazard ratio.** Heavy central lines represent the estimated adjusted hazard ratios, with shaded ribbons denoting 95% confidence intervals.

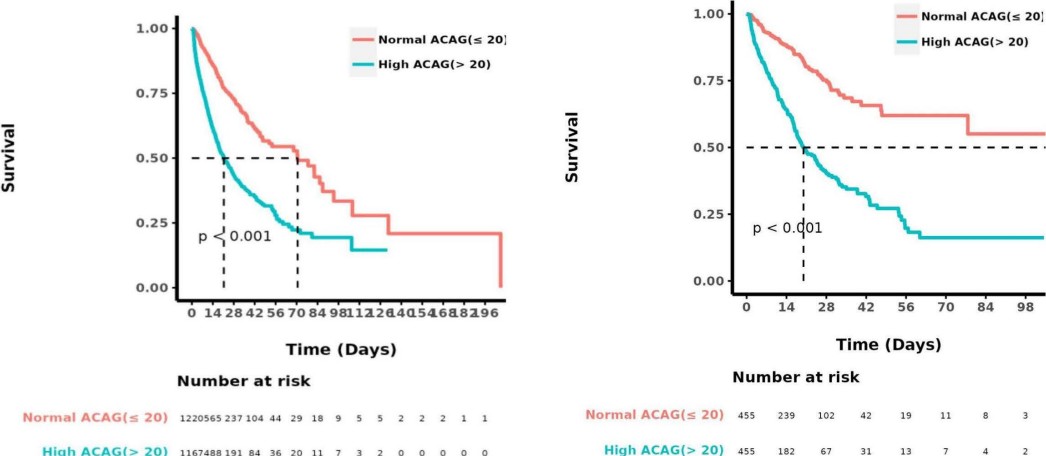

**Fig 3. Kaplan-Meier survival curve of cumulative survival rate during hospitalization for the normal-and high-ACAG groups before PSM and after PSM.**

this research revealed that the best threshold of ACAG for forecasting in-hospital mortality in individuals with Sepsis-AKI was 20.89 mmol/L. Conversely, the RCS plot showed a significant rise in mortality risk after the ACAG levels exceeded 20.28 mmol/L. To enhance clinical usefulness, a threshold of 20.0 mmol/L could be deemed suitable for assessing the risk of mortality in patients, ensuring a balance between precision and feasibility. The evidence indicates that high levels of ACAG (>20 mmol/L) at the start of continuous renal replacement therapy (CRRT) was linked to increased ICU mortality in AKI patients undergoing CRRT [27]. ACAG can therefore be used as an early warning sign of negative outcomes in these patients with Sepsis-AKI.

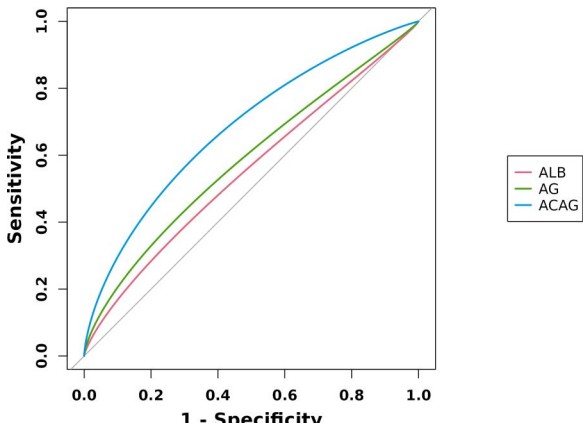

**Fig 4. ROC curves of ALB, AG, and ACAG (after PSM).** Abbreviation:ALB: albumin;AG:anion;ACAG:Albumin Corrected Anion Gap.

Table 4. Comparison of ROC curves(after PSM).

| Factor | AUC | Optimal cut-off | Youden's index | Sensitivity | Specificity |
|---|---|---|---|---|---|
| ALB | 0.551 | 2.558 | 0.1213 | 46.1% | 66.0% |
| AG | 0.581 | 19 | 0.1379 | 40.1% | 73.7% |
| ACAG | 0.697 | 20.89 | 0.3149 | 63.8% | 67.7% |

ALB:Albumin; AG:Anion gap; ACAG:Albumin Corrected Anion G

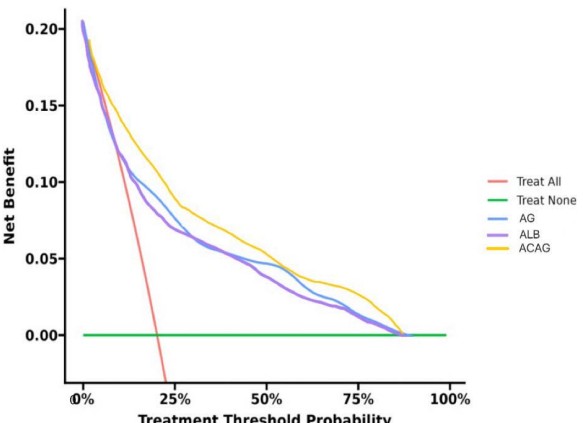

**Fig 5. Decision curve analysis (DCA) of albumin, AG, and ACAG (after PSM).** The preferred prognostic factor is also ACAG, the net benefit of which was the largest among the 3 factors.

Prior studies have shown that serum albumin is an autonomous predictor of mortality in elderly patients with sepsis [14]. Moreover, some investigation emphasized the importance of hypoalbuminemia as a critical factor in predicting the outcome of sepsis [28, 29]. ACAG combines albumin and AG to reflect the levels of both factors. Clinically, ACAG can indicate hypoalbuminemia and metabolic acidosis. Patients with sepsis commonly experience hypoalbuminemia due to inflammation or nutritional status, as well as acid-base disturbances, particularly in septic shock patients [30]. ACAG

elevation may prompt clinicians to address reversible causes of acidosis, such as hypovolemia, toxin accumulation, or mitochondrial dysfunction. For example, early CRRT initiation in high-ACAG patients could mitigate acid-base disturbances and inflammatory cascades, as suggested by Gaudry et al. (2016) [30]. Additionally, albumin supplementation might partially correct anion gap abnormalities in hypoalbuminemic patients [12, 28], though its clinical benefit remains debated [29]. Therefore, using ACAG to predict in-hospital mortality in patients with Sepsis-AKI holds significant predictive value. Compared with the normal-ACAG group, our study found that the high-ACAG group exhibited more pronounced renal function impairment, a greater prevalence of AKI stage 3, which suggested ACAG may predict the extent of kidney damage in patients. Patients with Sepsis-AKI are more likely to develop acid-base metabolic imbalance and are accompanied by hypoalbuminemia, ACAG can be a new and easily acquired indicator that can provide new ideas for clinical practice, so that doctors can judge the prognosis of patients more accurately. ACAG>20 may serve as a rapid, cost-effective biomarker to identify high-risk Sepsis-AKI patients requiring prioritized ICU admission or intensified monitoring. Similar to lactate-guided triage in sepsis [23], elevated ACAG reflects underlying metabolic acidosis and tissue hypoperfusion, which are hallmarks of organ dysfunction [17]. For instance, Zhong et al. (2022) reported that ACAG>18 independently predicted ICU mortality in CRRT-treated AKI patients, supporting its utility in resource allocation, integrating ACAG with existing scores (e.g., SOFA or KDIGO stages) could improve prognostic accuracy, as shown in AKI patients requiring continuous renal replacement therapy (CRRT) [15].

However, our study also had several limitations. Firstly, as this study was retrospective in nature, it was unable to definitively establish causality. Despite the use of multivariate adjustment and propensity matching score analysis, there is still a possibility of residual confounding factors influencing the clinical outcomes. Secondly, it is important to note that the ACAG was not continuously monitored throughout the study,our investigation solely focused on evaluating the prognostic value of the baseline ACAG for patients with Sepsis-AKI, disregarding any dynamic changes in the ACAG. Thirdly, due to the limitations of the MIMIC-IV database,we were unable to obtain the specific times at which albumin and AG were measured. Additionally, certain confounding factors such as Acute Physiology and Chronic Health Evaluation II (APACHE II) were not thoroughly taken into account, which could potentially influence the findings. Finally,this study had a sample size of moderate magnitude, a relatively short follow-up duration, and was solely conducted at a single center, potentially introducing selection bias. Therefore, more prospective studies with a larger cohort of subjects are needed to support the present findings.

## Conclusions

ACAG demonstrates the greatest predictive accuracy for in-hospital mortality among patients with Sepsis-AKI, outperforming both albumin and AG. Elevated serum ACAG (>20 mmol/L) is an independent risk factor for all-cause hospital mortality in patients with Sepsis-AKI. ACAG can be a new and easily acquired indicator that can provide new ideas for clinical practice.

## Author contributions

**Conceptualization:** Jian Liao.

**Data curation:** Dingyu Lu.

**Formal analysis:** Xiao Xiao.

**Investigation:** Dingyu Lu.

**Methodology:** Jian Liao.

**Resources:** Xiao Xiao.

**Supervision:** Wei Huang.

**Validation:** Xiao Xiao.

**Writing – original draft:** Jian Liao.

**Writing – review & editing:** Maojuan Wang.

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
