## [Decision Letter · Decision Letter 0]

Dear Dr. Liao,

Thank you for submitting your manuscript to PLOS ONE. After careful consideration, we feel that it has merit but does not fully meet PLOS ONE’s publication criteria as it currently stands. Therefore, we invite you to submit a revised version of the manuscript that addresses the points raised during the review process.

We look forward to receiving your revised manuscript.

Kind regards,

Ahmet Murt

Academic Editor

PLOS ONE

Journal Requirements:

-https://doi.org/10.3389/fcvm.2023.1099003

-DOI: 10.1016/j.cca.2021.07.021

-https://doi.org/10.1186/s12933-023-01864-x

In your revision ensure you cite all your sources (including your own works), and quote or rephrase any duplicated text outside the methods section. Further consideration is dependent on these concerns being addressed.

3. In the online submission form, you indicated that “Data can be acquired from corresponding author.”

 All PLOS journals now require all data underlying the findings described in their manuscript to be freely available to other researchers, either 1. In a public repository, 2. Within the manuscript itself, or 3. Uploaded as supplementary information. This policy applies to all data except where public deposition would breach compliance with the protocol approved by your research ethics board. If your data cannot be made publicly available for ethical or legal reasons (e.g., public availability would compromise patient privacy), please explain your reasons on resubmission and your exemption request will be escalated for approval.

Additional Editor Comments:

In this manuscript where authors analyzed the role of albumin corrected anion gap in sepsis related AKI, authors should specifically discuss what albumin correction will add in following patients who already have high anion gap. The utility of albumin correction should be given.

Also, I recommend that you check your data. E.g. Decimals of Na and BMI were not given and albumin levels are all same for groups in table 3. Reliability of data is highly crucial as our publication criteria.

Reviewers' comments:

Reviewer's Responses to Questions

**Comments to the Author**

1. Is the manuscript technically sound, and do the data support the conclusions?

Reviewer #1: Partly

Reviewer #2: Yes

2. Has the statistical analysis been performed appropriately and rigorously?

Reviewer #1: Yes

Reviewer #2: Yes

3. Have the authors made all data underlying the findings in their manuscript fully available?

Reviewer #1: No

Reviewer #2: Yes

4. Is the manuscript presented in an intelligible fashion and written in standard English?

Reviewer #1: Yes

Reviewer #2: Yes

Reviewer #1: Dear Authors,

I have received your manuscript entitled "The role of albumin-corrected anion gap as a predictor of all-cause mortality in patients with Sepsis-AKI: a propensity score-matched cohort study." However, I would like to bring to your attention some important points that need to be addressed in your manuscript.

Major Issues:

1. There is no data provided on causes of acute kidney injury (AKI) such as drugs, toxins, and other factors. These data should be included, and you should statistically analyze these contributing factors.

2. How did you classify acute kidney injury? You need to explain the criteria related to AKI according to the KDIGO guidelines.

3. Please provide details on how patients were diagnosed with AKI stages I, II, and III. How were the patients classified into these AKI categories?

4. The definition of sepsis needs to be more clearly defined. The sentence "Sepsis is a syndrome characterized by life-threatening organ dysfunction and is a frequently encountered serious complication of critical illnesses such as trauma, infection, and shock" should be more comprehensive.

5. For the inclusion criteria, you wrote "patients with Sepsis-induced AKI (SOFA score ≥ 2 and also suspected of infection) during ICU admission." Is this the correct definition of sepsis-induced AKI? I am not familiar with this terminology in the literature. Please review this point.

Best regards.

Reviewer #2: 1- Operational definition of sepsis AKI need to be mentioned details in methodology

2- Do you think table before and after PSM analysis required.

3- To add more literature on discussion about similar findings in other cohort such as shock. pancreatitis, trauma, or any other causes of well-known metabolic acidosis

**Do you want your identity to be public for this peer review?** For information about this choice, including consent withdrawal, please see our Privacy Policy

Reviewer #1: **Yes: ** Fethi Gül

Reviewer #2: No

---

## [Author Response · Author response to Decision Letter 1]

16 Oct 2024

Dear Editor and reviewers,

Thank you for reviewing our manuscript and for the constructive comments, which greatly helped us to improve the manuscript (PONE-D-24-27578). We appreciate the promoting comments to our study, and we have accepted and revised as recommended in this revised manuscript. We highlighted all the revisions in yellow/red colour.

We would like to express our great appreciation to you and reviewers for comments on our paper. Looking forward to hearing from you.

Thank you and best regards. 

The main corrections in the paper and the responds to the comments are as following:

Additional Editor Comments: In this manuscript where authors analyzed the role of albumin corrected anion gap in sepsis related AKI, authors should specifically discuss what albumin correction will add in following patients who already have high anion gap. The utility of albumin correction should be given.Also, I recommend that you check your data. E.g. Decimals of Na and BMI were not given and albumin levels are all same for groups in table 3. Reliability of data is highly crucial as our publication criteria.

Response: 1、Thanks to the editor’comment. A high AG is often associated with severe clinical prognosis, especially in patients with sepsis. Patients with sepsis often have hyperlactatemia and metabolic acidosis, resulting in increased AG, but patients with sepsis are often accompanied by hypoalbuminemia. Since albumin is negatively charged that will lead to a decrease in AG,hypoalbuminemia may lead to false negative AG results, affecting the accurate judgment of the results. Therefore, the ACAG was used to more accurately reflect the presence of unmeasured anions by adjusting AG based on albumin levels. We have supplemented this in the Discussion section.

2、Na in MIMIC IV is an integer. We conduct statistical analysis again and correct the contents of the table. In actual operation, even after PSM, it is possible that the matched groups still have differences in some indicators, or are exactly the same. This depends on the characteristics of the data itself, the matching algorithm chosen (such as nearest neighbor matching, caliper matching, etc.), the stringency of the matching (such as the size of the caliper value), and the size of the sample size. If the matched two groups are exactly the same on a certain metric, this may mean that the two groups themselves are similar on that metric, or that the matching algorithm was very efficient at finding similar control groups.

Reviewer #1:

Comment 1:There is no data provided on causes of acute kidney injury (AKI) such as drugs, toxins, and other factors. These data should be included, and you should statistically analyze these contributing factors.

Response: I would like to extend my gratitude for your valuable feedback on our manuscript. Your suggestion regarding the inclusion of data on the causes of AKI, such as drugs, toxins, and other contributing factors, is indeed insightful.We recognize that incorporating data on the specific etiologies leading to AKI would enhance the depth and precision of our analysis. However, our study relied on the MIMIC-IV database, a publicly accessible and widely utilized clinical database that, while extensive, does not encompass all possible clinical details, including specific causes of AKI like particular medications or toxin exposures. Our choice of the MIMIC-IV database was driven by the breadth and diversity of the data it offers, allowing us to investigate general trends and outcomes in AKI, which is crucial for understanding the disease's impact and improving patient prognosis. Although our study has its limitations, we believe that the analysis of the relationship between the ACAG and AKI prognosis provides clinicians with a useful prognostic tool.

We concur that future research should consider including these data. We recommend that subsequent studies might utilize other databases that contain more detailed information or conduct prospective studies designed to collect such data. We are also open to exploring these factors in our future research endeavors.Once again, we appreciate your suggestions and look forward to further discussions, hoping to address your concerns adequately.

Comment 2: How did you classify acute kidney injury? You need to explain the criteria related to AKI according to the KDIGO guidelines.

Response: Thanks for the reviewer’s comment. The definition criteria of AKI have been supplemented according to KDIGO (references 19). We have mentioned this in the part of Methods marked in yellow.

Comment 3: Please provide details on how patients were diagnosed with AKI stages I, II, and III. How were the patients classified into these AKI categories?

Response: When extracting information on patients diagnosed with AKI from the MIMIC IV database, the stage of AKI was included. Perform a literature search using “MIMIC IV” as the keyword and we can find that many studies have extracted information on AKI and staging. We have made corresponding additions to make the article more complete.

Comment 4: The definition of sepsis needs to be more clearly defined. The sentence "Sepsis is a syndrome characterized by life-threatening organ dysfunction and is a frequently encountered serious complication of critical illnesses such as trauma, infection, and shock" should be more comprehensive.

Response: Thanks to the reviewers’comments, we have refined the definition of sepsis in the Introduction section.

Comment 5: For the inclusion criteria, you wrote "patients with Sepsis-induced AKI (SOFA score ≥ 2 and also suspected of infection) during ICU admission." Is this the correct definition of sepsis-induced AKI? I am not familiar with this terminology in the literature. Please review this point.

Response: Thank you for your insightful comments and for drawing our attention to the need for a more detailed definition of Sepsis-Associated AKI in our manuscript. In response to your comment, we have now included a more detailed explanation of Sepsis-AKI in the Methods section of our manuscript. We have referenced the Kidney Disease: Improving Global Outcomes (KDIGO) clinical practice guidelines.The guideline defined AKI as an abrupt decrease in kidney function occurring over 7 days or fewer: increase in SCr by ≥ 50% within 7 d, or increase in SCr by ≥ 0.3 mg/dL ( ≥26.5 mmol/l) within 48 h, or oliguria.

Reviewer #2:

Comment 1: Operational definition of sepsis AKI need to be mentioned details in methodology.

Response: Dear Reviewer,Thank you for your valuable feedback and for pointing out the need to clarify the definition of Sepsis-AKI in our manuscript.In line with the comments made by reviewer#1, we have added the definition of Sepsis-AKI in the methods section.

Comment 2: Do you think table before and after PSM analysis required.

Response: We appreciate your question regarding the necessity of including tables that display the analysis before and after Propensity Score Matching (PSM). We believe that presenting both tables serves multiple important purposes in our study.1.Transparency: Showing the data before PSM allows readers to see the original distribution of variables and any potential confounding factors that might exist between groups.2.Assessment of PSM Effectiveness: The comparison of the tables before and after PSM demonstrates the effectiveness of the matching process in balancing covariates between groups. This helps in validating the robustness of our results.3.Context and Interpretation: The before-and-after presentation provides context for the changes that occur as a result of the PSM, allowing readers to interpret the impact of the matching on the study outcomes.4.Completeness: Including both tables ensures that our report is comprehensive and thorough, which is particularly important for studies utilizing complex statistical methods like PSM. We have carefully considered an alternative approach where we might only present the PSM results, but we concluded that the full presentation better serves the readers by providing a complete picture of the study's methodology and findings.

Comment 3: To add more literature on discussion about similar findings in other cohort such as shock. pancreatitis, trauma, or any other causes of well-known metabolic acidosis.

Response: We have added References 23 and 24 in the Discussion

section.

Dear Editor and reviewers,We hope that the revision is acceptable, and your favorable consideration of our manuscript is greatly appreciated.

Best regards.

Jian Liao

2024-10-07

---

## [Decision Letter · Decision Letter 1]

Dear Dr. Liao,

Thank you for submitting your manuscript to PLOS ONE. After careful consideration, we feel that it has merit but does not fully meet PLOS ONE’s publication criteria as it currently stands. Therefore, we invite you to submit a revised version of the manuscript that addresses the points raised during the review process.

We look forward to receiving your revised manuscript.

Kind regards,

Ahmet Murt

Academic Editor

PLOS ONE

Additional Editor Comments:

Our impartial reviewers raised concerns for further revisions of your article. Please find their comments enclosed.

Also, I recommend that you check your data. E.g. Decimals of Na and BMI were not given and albumin levels are all same for groups in table 3. Reliability of data is highly crucial as our publication criteria.

Reviewers' comments:

Reviewer's Responses to Questions

**Comments to the Author**

Reviewer #2: All comments have been addressed

Reviewer #3: (No Response)

2. Is the manuscript technically sound, and do the data support the conclusions?

Reviewer #2: Yes

Reviewer #3: Yes

3. Has the statistical analysis been performed appropriately and rigorously?

Reviewer #2: Yes

Reviewer #3: Yes

4. Have the authors made all data underlying the findings in their manuscript fully available?

Reviewer #2: Yes

Reviewer #3: Yes

5. Is the manuscript presented in an intelligible fashion and written in standard English?

Reviewer #2: Yes

Reviewer #3: Yes

Reviewer #2: 1-Well done and congratulations.

2-Recheck the tables: Just make sure the table format in line with journal format.

3-Recheck the citations and references: To make sure reference also in line with the journal format.

Reviewer #3: The author has given a thoughtful revision and addressed all the previous reviewer comments. While the revised version is very much improved, I would like to offer additional suggestions that may further strengthen the manuscript if it is accepted for publication.

1) I’ve noticed that some of the sections, particularly in the methodology and result sections, exhibit typographical errors, incorrect punctuation and lack of fluency which may affect clarity. I recommended a careful proofreading to ensure precision in the manuscript information.

2) I suggest removing ‘the following reports provide further insights’ from the last statement in the final paragraph of the introduction section, as the arguments are already adequate.

3) Could the author clarify the Sepsis-3 definition in the paragraph? Is it referring to the definition of Sepsis-AKI or sepsis in general?

4) Please rewrite the definition of KDIGO and the ethical part in a fluent manner.

5) What does ‘exclusively acquired laboratory variables’ mean? Please rephrase, in order to add to a better understanding.

6) In the statistical part, may the author explain the reason of choosing p-value < 0.05 instead of p < 0.2 for multivariable Cox hazard regression? By choosing a p-value < 0.2, the author may ensure there is no important variable missing. PSM may eliminate cofounding by pairing similar characteristics but does not replace potential confounders that may be prematurely excluded.

7) May I know how the author checks for the balance of the data between matched cohorts?

8) For the result section, I would like to suggest a clear display of data by grouping the explanation on data pertaining to Table 1 and Table 3. Does Table 3 referring to the PSM data? Currently the titles of Table 1 and Table 3 are the same with similar variables. I also did not appreciate the explanation of Table 3 information, or any explanation comparing before and after cohort matching of variables.

9) For the predictors, I would like to suggest that extracting the independent variables before and after PSM in a table, and kindly highlight any univariate variables that change in terms of significance after PSM. It is important to have data transparency, however data organisation is also vital to the reader. One way to address these is to submit the complete information of Table 2 and 4 as supplementary data.

10) I would like to suggest the author add a depth of discussion with literature evidence in the final part of the discussion. Now we know that cAG>20 predicts all-cause mortality for sepsis-AKI patients. How do you relate it with clinical significance? As an example, will it helps in patient triaging? Will we now adopt a different approach in this population?

**Do you want your identity to be public for this peer review?** For information about this choice, including consent withdrawal, please see our Privacy Policy

Reviewer #2: No

Reviewer #3: **Yes: ** Aliza Mohamad Yusof

---

## [Author Response · Author response to Decision Letter 2]

25 Apr 2025

Reviewer #1:

Comment 2: Recheck the tables: Just make sure the table format in line with journal format.

Response: We have readjusted the format of the table.

Comment 3: Recheck the citations and references: To make sure reference also in line with the journal format.

Response: We have readjusted the format of the references.

Reviewer #2:

Comment 1: I’ve noticed that some of the sections, particularly in the methodology and result sections, exhibit typographical errors, incorrect punctuation and lack of fluency which may affect clarity. I recommended a careful proofreading to ensure precision in the manuscript information.

Response: We have revised and polished the language in the Methods and Results sections to enhance the accuracy of the manuscript.

Comment 2: I suggest removing ‘the following reports provide further insights’ from the last statement in the final paragraph of the introduction section, as the arguments are already adequate.

Response: We have removed this sentence.

Comment 3: Could the author clarify the Sepsis-3 definition in the paragraph? Is it referring to the definition of Sepsis-AKI or sepsis in general?

Response: In the introduction section, we defined Sepsis-3, and in the methodology section, we defined Sepsis-AKI, citing relevant guidelines or literature.

Comment 4: Please rewrite the definition of KDIGO and the ethical part in a fluent manner.

Response: We have made readjustments in “study population” section and highlighted them in yellow.

Comment 5: What does ‘exclusively acquired laboratory variables’ mean? Please rephrase, in order to add to a better understanding.

Response: We have made revisions again.

Comment 6: In the statistical part, may the author explain the reason of choosing p-value < 0.05 instead of p < 0.2 for multivariable Cox hazard regression? By choosing a p-value < 0.2, the author may ensure there is no important variable missing. PSM may eliminate cofounding by pairing similar characteristics but does not replace potential confounders that may be prematurely excluded.

Response: We sincerely appreciate the reviewers' valuable comments on our statistical section. We choosed p-value < 0.05 instead of p < 0.2 for multivariable Cox hazard regression due to the following reasons:

Avoid Overfitting: Retaining variables with p<0.2 in multivariable analysis might introduce noise and overfitting, compromising clinical interpretability and generalizability.

Integration of Clinical Relevance: In the final model, only variables with p < 0.05 were retained, and adjustments were made based on clinical expertise (such as known risk factors) to ensure that the selected variables were both statistically significant and biologically plausible.

Meanwhile, we have organized the data in Table 2 and Table 4 to avoid excessive data affecting readability.

Comment 7: May I know how the author checks for the balance of the data between matched cohorts?

Response: Standardized mean difference (SMD) values can also be used to assess the success of matching. Typically, an SMD of less than 0.1 for each variable is considered indicative of good balance in the cohort. We used P>0.05 to indicate that there was no statistical significance in other variables between the two groups, while the outcome measures were statistically significant, making it easier to understand and more readable.

We have provided additional explanations in the Results section, which have been highlighted in yellow.

Comment 8: For the result section, I would like to suggest a clear display of data by grouping the explanation on data pertaining to Table 1 and Table 3. Does Table 3 referring to the PSM data? Currently the titles of Table 1 and Table 3 are the same with similar variables. I also did not appreciate the explanation of Table 3 information, or any explanation comparing before and after cohort matching of variables.

Response: We appreciate the editor's comments on the inappropriate and unscientific presentation of the tables. We have merged Table 1 and 3 and newly included the SMD values, which allows for a more intuitive visualization of the differences in data before and after matching.The results section has been rephrased.

Comment 9:For the predictors, I would like to suggest that extracting the independent variables before and after PSM in a table, and kindly highlight any univariate variables that change in terms of significance after PSM. It is important to have data transparency, however data organisation is also vital to the reader. One way to address these is to submit the complete information of Table 2 and 4 as supplementary data.

Response: Thank you for the reviewers' suggestions. We have resubmitted the matched data as supplementary data.

Comment 10: I would like to suggest the author add a depth of discussion with literature evidence in the final part of the discussion. Now we know that ACAG>20 predicts all-cause mortality for sepsis-AKI patients. How do you relate it with clinical significance? As an example, will it helps in patient triaging? Will we now adopt a different approach in this population?

Response: Thank you for the reviewer's comments. We have supplemented the discussion section accordingly and highlighted the additions in yellow.

---

## [Decision Letter · Decision Letter 2]

The role of albumin-corrected anion gap as a predictor of all-cause mortality in patients with Sepsis-AKI: a propensity score-matched cohort study

PONE-D-24-27578R2

Dear Dr. Liao,

We’re pleased to inform you that your manuscript has been judged scientifically suitable for publication and will be formally accepted for publication once it meets all outstanding technical requirements.

Kind regards,

Ahmet Murt

Academic Editor

PLOS ONE

Additional Editor Comments (optional):

There are no additional comments

Reviewers' comments:

Reviewer's Responses to Questions

**Comments to the Author**

Reviewer #2: All comments have been addressed

Reviewer #3: All comments have been addressed

2. Is the manuscript technically sound, and do the data support the conclusions?

Reviewer #2: Yes

Reviewer #3: Yes

3. Has the statistical analysis been performed appropriately and rigorously?

Reviewer #2: Yes

Reviewer #3: Yes

4. Have the authors made all data underlying the findings in their manuscript fully available?

Reviewer #2: Yes

Reviewer #3: Yes

5. Is the manuscript presented in an intelligible fashion and written in standard English?

Reviewer #2: Yes

Reviewer #3: Yes

Reviewer #2: ACAG in abstract, do we need to spell it first full meaning because PSM was mentioned full followed by abbreviation. No standardization. ACAG full meaning was mentioned in introduction. It would be better in abstract as well similar like PSM.

Reviewer #3: Thank you for the thorough revisions. Besides the inconsistency in table numbering, my previous comments have been addressed satisfactorily.

**Do you want your identity to be public for this peer review?** For information about this choice, including consent withdrawal, please see our Privacy Policy

Reviewer #2: No

Reviewer #3: **Yes: ** Aliza Mohamad Yusof

---

## [Editor Report · Acceptance letter]

PONE-D-24-27578R2

PLOS ONE

Dear Dr. Liao,

I'm pleased to inform you that your manuscript has been deemed suitable for publication in PLOS ONE. Congratulations! Your manuscript is now being handed over to our production team.

Kind regards,

on behalf of

Dr. Ahmet Murt

Academic Editor

PLOS ONE